**Data Availability Statement:** Location of the data is in Dataverse.nl2. The DOI/accession number of each dataset is https://doi.org/10.34894/DLX0ZB.

# Understanding cervical cancer awareness in hard-to-reach areas of Bangladesh: A cross-sectional study involving women and household decisionmakers

Naheed Nazrul[1]*, Marlieke de Fouw[2©], Jogchum J. Beltman[2©], Janine de Zeeuw[3], Jurjen van der Schans[3], Jaap Koot[3], Kazi Golam Rasul[1], Mosamat Umma Kulsum[1], Md Shahnur Ahmed[1©], Anika Rahman[4], Aminur Rahman[4], Kazi Maruful Islam[5], Ashrafun Nessa[6], Christine Campbell[7], Jelle Stekelenburg[3,8]

1 Health Sector, Friendship NGO, Dhaka, Bangladesh, 2 Department of Gynecology, Leiden University Medical Centre, Leiden University, Leiden, Netherlands, 3 Department of Health Sciences, Global Health Unit, University Medical Center Groningen, University of Groningen, Groningen, Netherlands, 4 Health System and Population Studies Division, icddr, b, Dhaka, Bangladesh, 5 Department of Development Studies, University of Dhaka, Dhaka, Bangladesh, 6 Department of Gynecological Oncology, Bangabandhu Sheikh Mujib Medical University, Dhaka, Bangladesh, 7 Usher Institute, University of Edinburgh, Edinburgh, Scotland, United Kingdom, 8 Department of Obstetrics and Gynecology, Medical Center Leeuwarden, Leeuwarden, Netherlands

© These authors contributed equally to this work.
* naheed@friendship.ngo

## Abstract

### Introduction

In Bangladesh, the uptake of cervical cancer screening is low. Lack of knowledge and understanding of symptoms and risk factors contributes to low screening uptake. The purpose of this study was to explore the knowledge of cervical cancer risk factors and symptoms and to measure the association with socio-demographic characteristics among women and household decisionmakers living in hard-to-reach areas of Bangladesh.

### Methods

A cross-sectional survey was conducted in five districts in Bangladesh among women aged between 30 and 60 years, their husbands, and their mothers-in-law from April to September 2022. Data were collected using a modified version of the validated AWACAN questionnaire tool. The significance level was considered at p-value <0.05 and odds ratios with 95% confidence.

### Results

Nearly 50% of participating women in hard-to-reach areas of Bangladesh and their family decisionmakers had low levels of knowledge of the risk factors and symptoms of cervical cancer. Only 20% of respondents in our survey knew about HPV, the most important risk factor for developing cervical cancer. Most respondents were familiar with the terminology

A direct link to freely access each data set is https://dataverse.nl/dataset.xhtml?persistentId=doi%3A10.34894%2FDLXOZB&version=DRAFT.

**Funding:** •NN,MF,JJB,JDZ,JVDS,JK,KGR,MUK, MSA,AR,AR, AN,CC,JS •European Union's Horizon 2020 research and innovation programme grant agreement No 964270 and from the Ministry of Science and Technology, Department of Biomedical Technology in India, grant No 13213, under the Global Alliance for Chronic Diseases. •The funders had no role in study design, data collection and analysis, decision to publish, or preparation of the manuscript.

**Competing interests:** The authors have declared that no competing interests exist.

of cervical cancer as a disease; however, approximately 40% of respondents did not know that not adhering to cervical cancer screening could be seen as a risk factor. Women do not make decisions about participation in cervical cancer screening on their own. Knowledge of cervical cancer risk factors and symptoms among decisionmakers was significantly associated with higher education and higher household monthly expenditure.

## Conclusion

Women, their husbands, and mothers-in-law in hard-to-reach areas of Bangladesh had limited knowledge about cervical cancer risk factors and symptoms. Engaging these key decision-makers in targeted health education is vital to improve screening uptake. Conduction of future research to identify and address screening barriers is also essential for effective prevention efforts.

## Introduction

Cervical cancer poses a great threat to women's health [1]. Globally, in 2020, 604,000 new cases were diagnosed, and 342,000 women died from this cancer. Nearly 90% of new diagnoses and deaths were reported in Low and Middle-Income Countries (LMICs) [2]. Almost all cervical cancer cases (more than 95%) are associated with high-risk human papillomavirus (hrHPV) and are largely preventable through HPV vaccination and organized screening and treatment programs [3–6]. The disease burden of cervical cancer in High-Income Countries (HICs) decreased significantly following the introduction of systematic screening and treatment programs and by establishing national vaccination programs against HPV [1, 7–9]. Due to limited screening and vaccination programs in LMICs, age-standardized cervical cancer mortality rates are still at least six-fold higher than in HICs [10], and this disparity is a direct result of resource inequities [11]. The World Health Organization (WHO) considers cervical cancer as a significant public health problem, and therefore, WHO released a global strategy in 2020 to accelerate the elimination of cervical cancer worldwide [12].

In Bangladesh (a LMIC), approximately 64 million women of reproductive age are at risk [3, 13] of developing cervical cancer. The estimated annual number of new cases of cervical cancer is around 8,000, and annual deaths are around 5,000 making it the third most common cause of female cancer mortality in Bangladesh [3]. Without intervention, it is predicted that 505,703 Bangladeshi women will die of cervical cancer between now and 2070 [14]. A national screening program in Bangladesh was launched in 2004, using visual inspection of the cervix with acetic acid (VIA). It started as a pilot program in 16 of the 64 districts in the country, scaling up to 44 districts at the end of 2007 [15] and gradually expanding to the remaining districts [16]. However, screening activities are still opportunistic [15] and cervical cancer screening participation remains low [17]. Therefore, it is important to understand the barriers and facilitators for participation in screening.

Several studies have examined barriers to screening participation in different countries and contexts [17–21]. One of the most important barriers described is lack of knowledge and poor understanding of the importance of screening in women, their husbands, and other family decisionmakers [20, 22]. In Bangladesh, their knowledge has not previously been studied in rural and hard-to-reach areas. The implementation research project PRESCRIP-TEC (Prevention and Screening Innovation Projects Towards Elimination of Cervical Cancer) introduced

HPV self-sampling and context-adapted awareness activities among selected populations in hard-to-reach areas to strengthen screening with HPV testing as primary triage tool [23].

In the present study, which is part of the PRESCRIP-TEC research project, we aimed to explore the knowledge of risk factors and symptoms of cervical cancer in women and household decisionmakers in hard-to-reach areas of Bangladesh. Secondly, we aimed to assess the association between socio-demographic characteristics and knowledge of risk factors and symptoms of women and household decisionmakers.

## Materials and methods

We describe a cross-sectional study carried out by the International Centre for Diarrheal Disease Research, Bangladesh (ICDDR, B) and Friendship, a local non-governmental organization (NGO) which focuses on the most hard-to-reach, unaddressed, and climate-impacted areas in Bangladesh.

### Study setting

The research was carried out from April 24, 2022 to September 30, 2022 prior to the introduction of hrHPV self-sampling in the PRESCRIP-TEC project in five districts of Bangladesh. In the four northern districts of Kurigram, Gaibandha, Bogura, and Sirajganj, a total of eight sub-districts located in the remote riverine nomadic islands known as chars were included. In the southern district of Satkhira, one sub-district in the coastal area was included. These regions are hard to reach, and their marginalized rural communities are susceptible to food insecurity, water scarcity, and natural disasters due to climate change [24, 25]. Since 2012, cervical cancer screening with VIA has been gradually introduced by Friendship in these sub-districts.

### Study population and sample

As per national guidelines for cervical cancer screening in Bangladesh, eligible women were between the ages of 30 and 60. In addition, we recruited their husbands and mothers-in-law, as these individuals are commonly regarded as the primary decisionmakers within the family unit. In Bangladesh, husbands hold significant decision-making power in family matters, including healthcare [26]. Women's participation in decision-making is limited, and they must discuss their plans with their husbands and mother-in-law before making any decisions [27, 28].

Only respondents who could understand the study information and answer questions in the Bengali language were included. In order to detect a minimal difference of 10% in awareness between different subpopulations, based on a mean awareness in rural Bangladesh of approximately 70% of cervical cancer risk factors and cervical cancer symptoms [17, 21] assuming a two-sided two-sample test with an alpha of 0.05, a power of 80%, and a non-response rate of 15% we calculated the minimal sample size for data collection to be 400 women, 400 husbands, and 400 mothers-in-law.

Convenience sampling was used to select the sub-districts in the Friendship catchment areas where the existing VIA screening program was ongoing. The local health team provided an updated list of eligible women, which was used as the sampling frame. We selected 100 households from each of the four districts in the Northern region and 200 households from the district in the Southern region by random sampling. We randomly picked one household and moved to every third household after following a systematic sampling technique. In total 220 households were excluded if any of the three members were absent or did not provide informed consent.

## Instrument and data collection

The study used a household survey at the community level, adapted from the African Women's Awareness of Cancer (AWACAN) tool (www.awacan.online) [29]. The original AWACAN tool investigates awareness of breast and cervical cancer symptoms, risk factors, lay beliefs, confidence in the appraisal, help-seeking behavior, and barriers to seeking healthcare among women. It has been demonstrated that the tool is reliable and valid for use in Sub-Saharan Africa [29, 30]. The original AWACAN questionnaire consists of 65 questions about breast and cervical cancer. The tool employs a combination of open-ended and closed-ended questions. As the PRESCRIP-TEC program focused on cervical cancer, only the questions of the AWACAN tool related to cervical cancer awareness were used in this study. The original AWACAN tool was developed for women; therefore, we adjusted questions for household decisionmakers. The questionnaire was contextualized for Bangladesh use by adjusting 48 questions for relevance to the local context (S1 Questionnaire). The ICDDR-B research team forward translated the AWACAN questionnaire into the Bengali language and verified for language and dialect accuracy, followed by pilot testing of the adapted questionnaire. The questionnaire was readjusted, and five questions about the uptake of screening were added, bringing the total number of questions to 53. Based on the AWACAN validation study, we used the median knowledge score as a cut-off value to differentiate between low and high knowledge scores [29].

The study was conducted by 43 data collectors of Friendship. These data collectors were experienced medical professionals. All questionnaire data were collected digitally using the KOBO Toolbox, a free and open-source platform for mobile data collection (www.kobotoolbox.org). Each data collector received two online and one in-person training session on the AWACAN tool, KOBO Toolbox, the study objectives, and an information session on cervical cancer, symptoms, and risk factors.

All data collectors spoke fluent Bengali, and male and female data collectors were assigned to interview male and female respondents, respectively, to address social norms and culture. Missing and inconsistent data were again rechecked with the respondents within seven days. During the data collection, each woman, husband, and mother-in-law was interviewed separately in their homes to protect their privacy and to maintain confidentiality.

## Statistical analysis

The demographic characteristics and knowledge of risk factors and symptoms were analyzed using descriptive statistics. Median and interquartile ranges were used to measure knowledge scores, categorized as low knowledge (below the median) and high knowledge (median and above). To check the association among independent variables, the variance inflation factor (VIF) was used. Multivariate logistic regression models were employed by adjusting age, education, expenditure, and occupation to determine the strength of the association between knowledge of risk factors, symptoms, and socio-demographic characteristics. A p-value of 0.05 was considered statistically significant, and odds ratios with 95% confidence intervals were reported. All statistical analyses were performed using Stata 17.0 (StataCorp.2021). Stata Statistical Software: Release 17. College Station, TX: Stata Corp LLC.

## Ethics

The study received ethical clearance from the Institutional Review Board of International Centre for Diarrheal Disease Research, Bangladesh (ICDDR, B) (approval number 21029). The respondents provided written consent before data collection using an Informed Consent Form

(ICF) in simple Bengali. The principal investigator (PI) and the research team had access to the data and informed consent for this study.

# Results

## Socio-demographic characteristics of the respondents

In total, 820 households were approached to participate, in 220 households one of the decisionsmakers was missing or household members did not provide informed consent. Therefore, 1800 respondents from 600 households were interviewed: 600 women, 600 husbands, and 600 mothers-in-law. The socio-demographic characteristics of the respondents are listed in Table 1. Most respondents reported Islam (92.3%) as their religion. The majority of women (56.5%, n = 339), husbands (64.3%, n = 386) and mothers-in-law (97.3%, n = 584) did not complete primary education. Nearly all women (94.7%, n = 568) and mothers-in-law (98.7%, n = 592) were housewives, while most husbands (62.5%, n = 375) were farmers, fishermen, or boatmen. The monthly family expenditure of almost all households (95.2–96.2%) lies below the average expenditure of 26,207 BDT (276 USD) in rural areas of Bangladesh [31]. The majority of the respondents had heard about cervical cancer; 6.2% (n = 37) of the women had received a cervical cancer screening; the majority of these women 64.9% (n = 24.)had been screened more than five years previously. Few women (3.3%, n = 20)made their own decisions regarding their own health, whilst most decisions related to women's health were made by their partner (40.7%, n = 244) or jointly with their partner or someone else in the family (54.4%, n = 326).

## Knowledge of risk factors and symptoms of cervical cancer

Table 2 illustrates the knowledge of cervical cancer risk factors among respondents who had heard of cervical cancer (74.3%, n = 1337). Most women (80.3%), husbands (91.1%), and mothers-in-law (77.5%) reported multiple sexual partners as being a risk factor for developing cervical cancer in women. Other risk factors that were frequently mentioned were early sexual debut (before the age of 17), and multiparity, and more than 60% of respondents mentioned lack of cervical cancer screening as a risk factor. However, HPV infection was recognized as a risk factor by only 20% of all three groups of respondents.

Table 3 illustrates the knowledge of symptoms of cervical cancer among all respondents (n = 1800). Among women, vaginal itching was the most frequently reported possible symptom (80.8%), followed by persistent smelly vaginal discharge, intermenstrual bleeding, and heavy periods. Furthermore, 76.0% of men (n = 456) and 81.3% of mothers-in-law (n = 488) reported persistent odorous vaginal discharge as a possible symptom, followed by vaginal itching and vaginal bleeding during and after sexual intercourse.

The composite median knowledge score of cervical cancer risk factors for all three groups was 6 (IQR 4–8 for women and husbands, IQR 3–8 for mothers-in-law). A cut-off of higher or equal to 6 is considered high knowledge of risk factors.

The median composite knowledge score of cervical cancer symptoms was 8 (IQR 4–10) for women and mothers-in-law. The median knowledge score for husbands was 7 (IQR 4–9). Higher or equal to 7 or 8 is considered high knowledge of symptoms (S1 Table).

## Association of knowledge with socio-demographic variables

Table 4 illustrates that husbands with primary (OR 1.94 95% CI: 1.19–3.15, p = 0.007) and secondary (OR 2.50 95% CI: 1.10–5.70, p = 0.029) education were more likely to be knowledgeable about risk factors compared to husbands who had below primary education. Similarly,

**Table 1. Socio-demographic characteristics of respondents (n = 1800).**

| Variables | Women 30–60 years (n = 600), n (%) | Household decisionmakers (n = 1200), n (%) | |
|---|---|---|---|
| | | Husband (n = 600), n (%) | Mother-in-law (n = 600), n (%) |
| Mean age (years) ± SD | 37±5.33 | 43±6.62 | 65±9.08 |
| **Religion** | | | |
| Islam | 554 (92.3) | 557 (92.8) | 555 (92.5) |
| Hinduism | 46 (7.7) | 43 (7.2) | 45 (7.5) |
| **Education** | | | |
| Below primary | 339 (56.5) | 386 (64.3) | 584 (97.3) |
| Primary | 225 (37.5) | 152 (25.3) | 16 (2.7) |
| Secondary | 19 (3.2) | 43 (7.2) | 0 |
| Higher secondary and above | 17 (2.8) | 19 (3.2) | 0 |
| **Occupation** | | | |
| Housewife | 568 (94.7) | 0 (0) | 592 (98.7) |
| Day labor | 11 (1.8) | 111 (18.5) | 5 (0.8) |
| Farmer/Fisherman/Boatman | 2 (0.3) | 375 (62.5) | 3 (0.5) |
| Business | 11 (1.8) | 80 (13.3) | 0 |
| Others | 8 (1.4) | 34 (5.7) | 0 |
| **Monthly family expenditure (BDT & USD)** | | | |
| 0–5000 (0–52 USD) | | 11 (1.8) | 51 (8.5) |
| 5001–10000 (52–105 USD) | | 348 (58.0) | 337 (56.2) |
| 10001–20000 (105–210 USD) | | 212 (35.3) | 189 (31.5) |
| >20000 (>210 USD) | | 29 (4.8) | 23 (3.8) |
| **Information on knowledge of cervical cancer and screening history** | | | |
| Ever heard of cervical cancer | 486 (81.0) | 406 (67.7) | 445 (74.2) |
| Ever tested for cervical cancer | 37 (6.2) | | |
| **Last tested for cervical cancer** | | | |
| >5 years ago | 24 (64.9) | | |
| 3–5 years ago | 2 (5.4) | | |
| 1–2 years ago | 7 (18.9) | | |
| <1 year ago | 4 (10.8) | | |
| **Decisions regarding women's health** | | | |
| Myself | 20 (3.3) | | |
| Partner | 244 (40.7) | | |
| Myself and partner jointly | 265 (44.2) | | |
| Someone else in household | 21 (3.5) | | |
| Myself and someone else jointly | 61 (10.2) | | |

Table 5 showed husbands with primary education (OR 2.51 95% CI: 1.65–3.79, p <0.001) and secondary education (OR 3.59 95% CI: 1.62–7.97, p = 0.002) were more likely to be knowledgeable about symptoms compared to husbands who had below primary education.

In mothers-in-law household expenditure of 10,001–20,000 BDT (105–210 USD) was associated with more knowledge of risk factors compared to mothers-in-law whose expenditure was 0–5000 BDT (0–52 USD) (OR 3.06 CI: 1.42–6.61, p = 0.004), illustrated in Table 4.

Table 5 shows that the knowledge of symptoms of cervical cancer was also higher in mothers-in -law with a household expenditure above 5000 BDT (52 USD), compared to those who had a household expenditure of 0–5000 BDT (0–52 USD).

**Table 2. Knowledge of cervical cancer risk factors.**

|  | Women (n = 486), n (%) | Husband (n = 406), n (%) | Mother-in-law (n = 445), n (%) |
|---|---|---|---|
|  | Yes (n, (%) | Yes (n, (%) | Yes (n, (%) |
| HPV infection | 102 (20.9) | 84 (20.7) | 89 (20.0) |
| HIV/AIDS | 161 (33.1) | 220 (54.2) | 143 (32.1) |
| Other STDs | 165 (33.9) | 121 (29.8) | 128 (28.8) |
| Birth control pills/family planning for over 5 years | 251 (51.7) | 161 (39.7) | 201 (45.2) |
| Unprotected sex | 308 (63.4) | 305 (75.1) | 249 (55.9) |
| Smoking any cigarettes | 299 (61.5) | 251 (61.8) | 236 (53.0) |
| Uncircumcised sexual partners | 139 (28.6) | 108 (26.6) | 141 (31.7) |
| Sex before the age of 17 | 377 (77.6) | 322 (79.3) | 351 (78.8) |
| Giving birth to three or more children | 337 (69.3) | 251 (61.8) | 300 (67.4) |
| Multiple sexual partners | 390 (80.3) | 370 (91.1) | 345 (77.5) |
| Skipping cervical cancer screenings | 310 (63.8) | 253 (62.3) | 292 (65.6) |

Among women in the screening target group we did not identify determinants of knowledge of symptoms and risk factors of cervical cancer.

## Discussion

This study assessed knowledge of risk factors and symptoms of cervical cancer and its associated factors among respondents living in hard-to-reach areas of Bangladesh. The primary finding was that the majority of women and household decisionmakers had heard about cervical cancer, and knowledge levels of cervical cancer symptoms and risk factors were comparable between women, husbands, and mothers-in-law despite a low level of formal education and monthly household expenditure below the national average. However, only 6.2% of women interviewed reported having participated in screening previously, while more than 60% of respondents identified not participating in cervical cancer screening as a risk factor for developing cervical cancer.

Our findings are similar to a study in Zimbabwe conducted in 2014, where nearly 65% of the respondents had heard of cervical cancer, but only 3.8% of women had undergone cervical

**Table 3. Knowledge of cervical cancer symptoms.**

|  | Women (n = 600), n (%) | Husband (n = 600), n (%) | Mother-in-law (n = 600), n (%) |
|---|---|---|---|
|  | Yes, n (%) | Yes, n (%) | Yes, n (%) |
| Intermenstrual bleeding | 425 (70.8) | 353 (58.8) | 403 (67.2) |
| Chronic back pain | 341 (56.8) | 256 (42.7) | 307 (51.2) |
| Persistent smelly vaginal discharge | 470 (78.3) | 456 (76.0) | 488 (81.3) |
| Discomfort or pain during sex | 387 (64.5) | 389 (64.8) | 366 (61.0) |
| Excess or heavy periods | 425 (70.8) | 408 (68.0) | 392 (65.3) |
| Persistent diarrhea | 110 (18.3) | 81 (13.5) | 119 (19.8) |
| Post-menopausal bleeding | 326 (54.3) | 237 (39.5) | 316 (52.7) |
| Chronic lower abdominal/pelvic pain | 410 (68.3) | 350 (58.3) | 384 (64.0) |
| Vaginal bleeding during or after sex | 415 (69.2) | 429 (71.5) | 399 (66.5) |
| Blood in urine or stool | 253 (42.2) | 278 (46.3) | 244 (40.7) |
| Unexplained weight loss | 229 (38.2) | 183 (30.5) | 236 (39.3) |
| Vaginal itching | 485 (80.8) | 449 (74.8) | 458 (76.3) |

**Table 4. Association between socio-demographic factors and knowledge on cervical cancer risk factors.**

| Socio-demographic factors | Women (n = 486) | | Husband (n = 405) | | Mother-in-law (n = 445) | |
|---|---|---|---|---|---|---|
| | Adjusted OR (95% CI) | p-value | Adjusted OR (95% CI) | p-value | Adjusted OR (95% CI) | p-value |
| **Age** (years) | | | | | | |
| 30–39 | Ref. | | Ref. | | | |
| 40–49 | 0.76 (0.49–1.19) | 0.231 | 1.24 (0.78–1.95) | 0.362 | Ref. | |
| 50–59 | 1.72 (0.50–5.85) | 0.389 | 1.63 (0.88–3.03) | 0.119 | 0.73 (0.21–2.54) | 0.619 |
| >59 | - | | 0.59 (0.05–6.85) | 0.673 | 0.4 (0.12–1.34) | 0.137 |
| **Education** | | | | | | |
| Below primary | Ref. | | Ref. | | Ref. | |
| Primary | 1.07 (0.71–1.61) | 0.75 | **1.94 (1.19–3.15)** | 0.007 | 2.36 (0.62–9.07) | 0.21 |
| Secondary | 1.06 (0.36–3.13) | 0.923 | **2.50 (1.10–5.70)** | 0.029 | - | |
| Higher secondary and above | 1.77 (0.56–5.60) | 0.333 | 2.63 (0.78–8.86) | 0.118 | - | |
| **Expenditure (BDT)** | | | | | | |
| 0–5000 | Ref. | | Ref. | | Ref. | |
| 5001–10000 | 0.88 (0.22–3.50) | 0.86 | 0.44 (0.1–1.84) | 0.26 | 1.91 (0.92–3.96) | 0.82 |
| 10001–20000 | 1.65 (0.41–6.69) | 0.483 | 0.62 (0.15–2.67) | 0.526 | **3.06 (1.42–6.61)** | **0.004** |
| 20001–30000 | 1.72 (0.32–9.26) | 0.527 | 2.03 (0.31–13.42) | 0.463 | 2.47 (0.77–7.92) | 0.129 |
| **Occupation** | | | | | | |
| Housewife | Ref. | | - | | Ref. | |
| Day labor | 0.81 (0.20–3.36) | 0.775 | Ref. | | 0.96 (0.21–4.45) | 0.963 |
| Farmer/ Fisherman /Boatman | - | - | 0.79 (0.46–1.35) | 0.389 | - | - |
| Business | 0.96 (0.26–3.57) | 0.955 | 0.62 (0.29–1.32) | 0.212 | - | - |
| Others | 1.13 (0.24–5.36) | 0.875 | 1.21 (0.45–3.30) | 0.706 | - | - |

*Significance level: p-value<0.05

cancer screening [32], again associated with lack of awareness and knowledge of cervical cancer [32]. Also, a study among tribal women in the Southern India conducted from 2014 to 2017 and reported that 82.9% of the participants had heard of cervical cancer, but none of the study women participated screening previously due to poor knowledge of cervical cancer [33]. Similarly, a study in India conducted in 2009 found that some level of knowledge of cervical cancer screening was essential for participating in screening [34], indicating that lack of knowledge on cervical cancer seems to be a barrier to participating in screening programs.

The most commonly identified risk factor for causing cervical cancer was multiple sexual partners, as stated by all three groups of respondents. A similar result was found in a study conducted in Zimbabwe and India, where most respondents recognized having multiple sexual partners as a risk factor for developing cervical cancer [32, 35]. However, only 20% of respondents in our survey knew about HPV, the most important risk factor for developing cervical cancer, which correlates with multiple sexual partners. The low awareness of HPV may be due to the lack of a national HPV vaccination program in Bangladesh.

Overall, approximately half of the respondents, according to the median knowledge score, could not correctly identify the risk factors and symptoms of cervical cancer, which could lead to patient delay and, as a result, presenting in the late stage of the disease, which will impact prognosis. A study in India showed that women with the correct knowledge and higher awareness of cervical cancer (acceptors) had better treatment seeking behavior than non-acceptors [36]. Other studies confirm that lack of awareness of risk factors and symptoms may delay the

**Table 5. Association between socio-demographic factors and knowledge on cervical cancer symptoms.**

| Socio-demographic factors | Women (n = 600) | | Husband (n = 600) | | Mother-in-law (n = 600) | |
|---|---|---|---|---|---|---|
| | Adjusted OR (95% CI) | p-value | Adjusted OR (95% CI) | p-value | Adjusted OR (95% CI) | p-value |
| **Age** (years) | | | | | | |
| 30–39 | Ref. | | Ref. | | | |
| 40–49 | 1.03 (0.69–1.54) | 0.868 | 1.14 (0.78–1.69) | 0.495 | Ref. | |
| 50–59 | 0.89 (0.37–2.19) | 0.805 | 1.02 (0.61–1.69) | 0.948 | 1.4 (0.44–4.49) | 0.573 |
| >59 | - | | 0.73 (0.16–3.34) | 0.688 | 0.47 (0.15–1.45) | 0.19 |
| **Education** | | | | | | |
| Below primary | Ref. | | Ref. | | Ref. | |
| Primary | 0.82 (0.57–1.18) | 0.293 | **2.51 (1.65–3.79)** | <0.001 | 3.57 (0.98–13) | 0.053 |
| Secondary | 2.01 (0.69–5.87) | 0.2 | **3.59 (1.62–7.97)** | 0.002 | - | |
| Higher secondary and above | 1.27 (0.4–4.05) | 0.681 | 1.59 (0.54–4.67) | 0.396 | - | |
| **Expenditure** (BDT) | | | | | | |
| 0–5000 | | | Ref. | | Ref. | |
| 5001–10000 | | | **0.16 (0.03–0.74)** | 0.019 | **2.7 (1.38–5.26)** | 0.004 |
| 10001–20000 | | | 0.23 (0.05–1.08) | 0.063 | **3.42 (1.7–6.88)** | 0.001 |
| 20001–30000 | | | 0.58 (0.1–3.53) | 0.558 | **3.36 (1.16–9.73)** | 0.026 |
| **Occupation** | | | | | | |
| Housewife | Ref. | | - | | Ref. | |
| Day labor | 1.55 (0.49–4.87) | 0.457 | Ref. | | 2.47 (0.48–12.78) | 0.281 |
| Farmer/ Fisherman /Boatman | - | - | 0.52 (0.33–0.81) | 0.004 | | |
| Business | 1.38 (0.38–5) | 0.62 | 0.47 (0.25–0.9) | 0.023 | | |
| Others | 6.86 (0.78–60.13) | 0.082 | 1.46 (0.56–3.86) | 0.441 | | |

*Significance level: p-value<0.05

diagnosis of cancer [21, 37]. In this study, vaginal itching was the most reported symptom, although not a symptom of cervical cancer, and most often related to other issues like candidiasis and STDs [38–40].

Knowledge of cervical cancer risk factors and symptoms among decisionmakers was significantly associated with their educational level and expenditure status in the study population. High knowledge was associated with higher education and higher monthly expenditure, but other factors might influence knowledge. This is in line with an earlier study in Bangladesh that showed that women from high-income families and those with higher education levels were more likely to be aware of cervical cancer than those from low-income families and those with lower education [21]. This finding suggests that socio-economic status plays an important role in knowledge and awareness of cervical cancer in Bangladesh [41].

We hypothesized that socio-demographic factors, especially religion, education, occupation, and monthly expenditure, might influence knowledge retention and cervical cancer screening uptake. We included husbands and mothers-in-law as decisionmakers to understand better their influence on the decision-making process regarding women's health. In rural Bangladeshi communities, patriarchal social norms frequently position husbands as the primary decisionmakers due to their status as family breadwinners [42]. In our study, more than 94% of the female respondents are housewives, which makes them financially dependent on their husbands, which in turn will reduce the likelihood of participating in cervical cancer screening. Only 3.3% of women took their healthcare-related decisions by themselves; the majority of women were influenced by their husbands or mothers-in-law. Several studies suggest that decisionmakers such as husbands and mothers-in-law play a significant role in

women's health-related decision making and health-seeking behavior in cervical cancer screening [22, 42, 43].

An Indian study found that limited knowledge and negative attitudes regarding cervical cancer and screening among women and their male partners changed significantly after receiving sexual health education and improved knowledge and attitude towards screening [20]. This suggests that health education on cervical cancer should not focus solely on women but also needs to include men and other household decisionmakers.

Another context-specific barrier that could be associated with less screening participation is religion. The majority of the respondents in our study had a Muslim background. Previous studies revealed that cervical cancer screening is viewed from a different perspective due to conservative cultural and religious beliefs, fear, pain, and embarrassment, as observed in a study of educated Muslim women in Dubai [44]. A survey conducted among Muslim women in Ghana found a knowledge gap regarding cervical cancer and very low screening participation related to Islamic modesty and devotion to the Islamic faith [45].

There are several strengths of this study. We made use of the validated AWACAN tool, which was adapted for the local context. Unique geographical regions were selected, and we incorporated the family decisionmakers (husbands and mothers-in-law) to recognize their cultural role in this study.

There are some limitations to this study. The AWACAN tool does not specifically address the preventative nature of cervical cancer screening and, therefore, might miss specific barriers to the uptake of a screening program. Also, the data collectors could have influenced socially desirable answers, as some were well known to study respondents since they also provide other healthcare services for the Friendship organization. This study was also limited to existing cervical cancer screening programs of Friendship catchment areas that focused on hard-to-reach communities only, which may not represent other areas of Bangladesh.

The results of this study suggest that policymakers and program implementers should include men and other family decisionmakers in cervical cancer screening awareness strategies to improve screening uptake. Women in remote areas of Bangladesh depend on other decisionsmakers in terms of health-seeking behavior; this requires a different innovative communication strategy to reach both men and women with relevant messages. Besides the importance of these innovative communication strategies for improving the uptake of the screening program, health education sessions focused on knowledge about hrHPV will also be key to the successful implementation of the National Vaccination Program against hrHPV in Bangladesh.

## Conclusion

The majority of respondents heard about cervical cancer, while cervical cancer screening uptake was low. Nearly 50% of participating women and their family decisionmakers had low levels of knowledge of the risk factors and symptoms of cervical cancer, and almost all women depend on their husbands or mother-in-law for decision making regarding their health. Therefore, increasing health education dissemination and ensuring the involvement of household decisionmakers must be prioritized to increase the screening uptake. Additionally, future research is needed to identify the barriers that influence the low participation of healthy asymptomatic women living in hard-to-reach areas of Bangladesh in cervical cancer screening.

## Supporting information

**S1 Table. Knowledge score of cervical cancer risk factors and symptoms.**
(DOCX)

**S1 Questionnaire. AWACAN questionnaire.**
(DOCX)

## Acknowledgments

Prevention and Screening Innovation Project–Towards Elimination of Cervical Cancer (PRE-SCRIP-TEC) is a research consortium project delivered through a collaboration of fifteen consortium members. This study could not have been completed without Friendship and ICDDR, B team effort. Their sincere contribution, hard work, and dedication completed this study on time. Last but not least, thanks to all the respondents in the hard-to-reach areas who participated in this research.

## Author Contributions

**Conceptualization:** Marlieke de Fouw, Jogchum J. Beltman, Janine de Zeeuw, Jurjen van der Schans, Jaap Koot, Anika Rahman, Aminur Rahman, Ashrafun Nessa, Christine Campbell, Jelle Stekelenburg.

**Data curation:** Anika Rahman, Aminur Rahman.

**Formal analysis:** Naheed Nazrul, Md Shahnur Ahmed, Anika Rahman, Aminur Rahman.

**Funding acquisition:** Naheed Nazrul, Jaap Koot, Aminur Rahman, Jelle Stekelenburg.

**Investigation:** Marlieke de Fouw, Janine de Zeeuw, Jurjen van der Schans, Jaap Koot, Aminur Rahman, Jelle Stekelenburg.

**Methodology:** Naheed Nazrul, Marlieke de Fouw, Jogchum J. Beltman, Janine de Zeeuw, Jurjen van der Schans, Jaap Koot, Md Shahnur Ahmed, Anika Rahman, Aminur Rahman, Christine Campbell, Jelle Stekelenburg.

**Project administration:** Naheed Nazrul, Jaap Koot, Md Shahnur Ahmed, Jelle Stekelenburg.

**Resources:** Naheed Nazrul, Kazi Golam Rasul, Mosamat Umma Kulsum.

**Software:** Md Shahnur Ahmed, Aminur Rahman.

**Supervision:** Naheed Nazrul, Marlieke de Fouw, Jogchum J. Beltman, Janine de Zeeuw, Jurjen van der Schans, Jaap Koot, Kazi Golam Rasul, Mosamat Umma Kulsum, Aminur Rahman, Christine Campbell, Jelle Stekelenburg.

**Validation:** Jaap Koot, Anika Rahman, Aminur Rahman, Jelle Stekelenburg.

**Visualization:** Janine de Zeeuw, Jurjen van der Schans, Jaap Koot, Md Shahnur Ahmed, Anika Rahman, Christine Campbell, Jelle Stekelenburg.

**Writing – original draft:** Naheed Nazrul, Md Shahnur Ahmed.

**Writing – review & editing:** Naheed Nazrul, Marlieke de Fouw, Jogchum J. Beltman, Janine de Zeeuw, Jurjen van der Schans, Jaap Koot, Kazi Golam Rasul, Mosamat Umma Kulsum, Md Shahnur Ahmed, Anika Rahman, Aminur Rahman, Kazi Maruful Islam, Ashrafun Nessa, Christine Campbell, Jelle Stekelenburg.

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
