## [Decision Letter · Decision Letter 0]

22 Jan 2024

PONE-D-23-29440Understanding cervical cancer awareness in hard-to-reach areas of Bangladesh: a cross-sectional study involving women and household decisionmakersPLOS ONE

Dear Dr. Nazrul,

Thank you for submitting your manuscript to PLOS ONE. After careful consideration, we feel that it has merit but does not fully meet PLOS ONE’s publication criteria as it currently stands. Therefore, we invite you to submit a revised version of the manuscript that addresses the points raised during the review process.

Please submit your revised manuscript by Mar 07 2024 11:59PM. If you will need more time than this to complete your revisions, please reply to this message or contact the journal office at plosone@plos.org. Please include the following items when submitting your revised manuscript:A rebuttal letter that responds to each point raised by the academic editor and reviewer(s). You should upload this letter as a separate file labeled 'Response to Reviewers'.A marked-up copy of your manuscript that highlights changes made to the original version. You should upload this as a separate file labeled 'Revised Manuscript with Track Changes'.An unmarked version of your revised paper without tracked changes. You should upload this as a separate file labeled 'Manuscript'.

We look forward to receiving your revised manuscript.

Kind regards,

Adu Appiah-Kubi, MBChB, CEMBA, FGCS

Academic Editor

PLOS ONE

Journal Requirements:

Reviewers' comments:

Reviewer's Responses to Questions

**Comments to the Author**

1. Is the manuscript technically sound, and do the data support the conclusions?

Reviewer #1: Yes

Reviewer #2: Partly

2. Has the statistical analysis been performed appropriately and rigorously? 

Reviewer #1: Yes

Reviewer #2: Yes

3. Have the authors made all data underlying the findings in their manuscript fully available?

Reviewer #1: No

Reviewer #2: Yes

4. Is the manuscript presented in an intelligible fashion and written in standard English?

Reviewer #1: Yes

Reviewer #2: Yes

5. Review Comments to the Author

Reviewer #1: Few issues required further revision:

Method:

1. The inclusion criteria appear to be unclear, particularly regarding the justification for including "husbands" in the study. Authors are encouraged to provide a more explicit rationale for this inclusion.

2. The decision to limit the study to women aged 30-60 raises questions. If this restriction aligns with national guidelines for cervical screening, it would be beneficial for the authors to elucidate. Additionally, considering a broader age range might enhance the study's generalizability.

3. The inclusion of 400 women, 400 husbands, and 400 household decision-makers needs clarification. The rationale behind separating these three groups should be justified.

4. The term "household decision-maker" lacks clarity and requires a more detailed definition. Authors are advised to provide explanation to avoid confusion, specifying whether it exclusively refers to husbands or includes others.

5. The study originally calculated a required sample size of 400 for each arm, but the final sample collected was 600. Authors should address this discrepancy and provide a clear explanation for the deviation from the initially determined sample size.

6. The statement that "a total of 1800 respondents were interviewed" raises concerns about its coherence. Authors are encouraged to revise and provide a more accurate representation of the study's participant recruitment and response rates.

Results:

1. The use of "family expenditure" rather than "household income" needs clarification. Authors should justify this choice to ensure a clear understanding of the financial metric employed.

2. Tables 4 and 5 present results that are causing confusion. Authors should elucidate the reasons for selecting specific parameters as references, such as age 30-39, education below primary, and expenditure 0-5000, to enhance the interpretability of the findings.

Reviewer #2: Reported study is a cross-sectional survey conducted in five districts in Bangladesh among women aged between 30 and 60 years, their husbands and their mothers-in-law from April to September 2022.

Lines 64-65: Need more clarity.

Results of multivariate analysis could be interpreted and reported in a better way to identify the determinants of good knowledge or identification of symptoms etc.

Discussion part please mention the other studies along with the year it was conducted.eg. the study conducted in India and Zimbawe

6. PLOS authors have the option to publish the peer review history of their article (what does this mean?). If published, this will include your full peer review and any attached files.

Reviewer #1: No

Reviewer #2: **Yes: **Gauravi Ashish Mishra

---

## [Author Response · Author response to Decision Letter 0]

12 Mar 2024

March 6, 2024

Response to comments raised by Editors and Reviewers on manuscript PONE-D-23-29440 titled: Understanding cervical cancer awareness in hard-to-reach areas of Bangladesh: a cross-sectional study involving women and household decisionmakers. 

Dear Editor

We appreciate the editors and reviewers for taking the time to review our manuscript PONE-D-23-29440 and thank them for the valuable comments they raised. Below are our responses to the comments in numerical order. We made adaptations to the manuscript according to the suggestions and attached the main manuscript, both with track changes and as a clean copy. 

Comments to the author

Reviewer#1

Comment 1: The inclusion criteria appear to be unclear, particularly regarding the justification for including "husbands" in the study. Authors are encouraged to provide a more explicit rationale for this inclusion.

Response 1: Thank you for the feedback. The explanation is given in the study population and sample section from lines 131 to 134, and we have explained that in Bangladesh, husbands hold significant decision-making power in family matters, including healthcare. Women's participation in decision-making is limited, and they must discuss their plans with their husbands and mother-in-law before making any decisions.

Comment 2: The decision to limit the study to women aged 30-60 raises questions. If this restriction aligns with national guidelines for cervical screening, it would be beneficial for the authors to elucidate. Additionally, considering a broader age range might enhance the study's generalizability.

Response 2: For ethical issues and to get permission from the Ministry of Health (MoH), study participants were kept at the existing screening policy age (30-60). Indeed, we selected the age group as outlined in the national guidelines, and we aimed to understand their knowledge and screening behavior and not as a representative of the population of Bangladesh. We selected this age group to use the results to contextualize the awareness strategy and uptake of screening before implementation. 

Comment 3: The inclusion of 400 women, 400 husbands, and 400 household decisionmakers needs clarification. The rationale behind separating these three groups should be justified.

Response 3: Thank you for the comment to clarify our inclusion criteria. Bangladesh is a patriarchal society where men are involved in household decision-making processes, including health-related issues. We refer to our response on comment 1 for background literature about decision-making and the involvement of husbands. Furthermore, mothers-in-law can play an important role in decision-making processes in families in Bangladesh. Therefore, we wanted to investigate cervical cancer awareness in all household decisionmakers, including the husband and mother-in-law, and their respective roles in decision-making, and not only women in the target group. Since it was not known at baseline who was the decisionmakers regarding cervical cancer screening, we decided to separate the 3 groups as potential decisionmakers in the household.

The minimal sample size for assessing the awareness in a specific population was calculated at 400 women and their husbands and mothers-in-law (as also discussed in the methodology section of the manuscript). For another research objective that was included in the baseline study of the PRESCRIP-TEC program we targeted 600 women (to assess coverage of cervical cancer screening at the start of the program). Therefore, we had the opportunity to survey more respondents than the sample size and since we included different areas of Bangladesh, including more participants would add to the generalizability of our study findings. 

Comment 4: The term "household decisionmaker" lacks clarity and requires a more detailed definition. Authors are advised to provide explanations to avoid confusion, specifying whether it exclusively refers to husbands or includes others.

Response 4: Thank you for this remark. In this manuscript, husbands and mothers-in-law were called household decisionmakers as the current power dynamics of the study population. Please find it in the response 1 above.

Comment 5: The study originally calculated a required sample size of 400 for each arm, but the final sample collected was 600. Authors should address this discrepancy and provide a clear explanation for the deviation from the initially determined sample size.

Response 5: Thank you for your feedback, we responded to your comment in our response on comment 3 above. 

Comment 6: The statement that "a total of 1800 respondents were interviewed" raises concerns about its coherence. Authors are encouraged to revise and provide a more accurate representation of the study's participant recruitment and response rates.

Response 6: For further clarification of the included number of participants, we refer to our response to comment 3 and 5. In the sampling strategy, we randomly selected households in the sub-districts by using a sampling frame of all households. We included households if woman, husband and mother-in-law were present and provided informed consent to participate in the study. In case one or more household decisionmakers were not present or did not provide informed consent, the neighboring household was approached to participate. With this sampling strategy we minimized selection bias. Overall, 820 households were approached to participate, and 220 households were excluded due to the absence of one of the household decisionmakers, or no informed consent was obtained. We added this to our manuscript from lines 201 to 204. 

Comment 7: The use of "family expenditure" rather than "household income" needs clarification. Authors should justify this choice to ensure a clear understanding of the financial metric employed.

Response 7: Using monthly expenditure instead of income to gauge wealth status offers several advantages. Expenditure more accurately reflects living standards and quality of life, as it accounts for non-monetary income and is less affected by income's seasonal fluctuations. It circumvents potential underreporting or over-reporting of income, common in rural economies with diverse and non-quantifiable income sources. Furthermore, expenditure data is often easier for respondents to recall and report, making it a more reliable indicator of long-term economic status. This approach is more culturally sensitive and more relevant for policy making, as it directly reflects the needs and priorities of households.

Comment 8: Tables 4 and 5 present results that are causing confusion. Authors should elucidate the reasons for selecting specific parameters as references, such as age 30-39, education below primary, and expenditure 0-5000, to enhance the interpretability of the findings.

Response 8: It is inherent to the method of logistic regression we selected for statistical analysis that these categories function as reference groups; due to the regression model used, this will not affect our findings of the strength of the associations we present. Selecting a different reference group would, therefore, not alter our outcome. 

Reviewer #2: 

Comment 1: Reported study is a cross-sectional survey conducted in five districts in Bangladesh among women aged between 30 and 60 years, their husbands, and their mothers-in-law from April to September 2022.

Lines 64-65: Need more clarity.

Response 1: Yes, thanks for pointing this out. We changed and revised from lines 62 to 65.

Comment 2: Results of multivariate analysis could be interpreted and reported in a better way to identify the determinants of good knowledge or identification of symptoms etc.

Response 2: Thank you for your feedback. We have adjusted the manuscript and present the determinants of good knowledge of risk factors and symptoms for women, husbands and mothers-in-law separately. We believe this will facilitate the interpretation of the results. 

Comment 3: Discussion part please mention the other studies along with the year it was conducted.eg. the study conducted in India and Zimbabwe.

Response 3: Thank you for this feedback. We have made the necessary adjustments in the discussion part from lines 276 to 284. We have included the published year for these two studies only, not for all the studies. We would like to focus on the results and the findings.

Thank you for taking the time to review our updated manuscript and feedback. We trust that the adaptations will make our manuscript more suitable for publication in your journal.

We are looking forward to your response.

Kind regards

Naheed Nazrul

Corresponding author

---

## [Decision Letter · Decision Letter 1]

13 May 2024

Understanding cervical cancer awareness in hard-to-reach areas of Bangladesh: a cross-sectional study involving women and household decisionmakers

PONE-D-23-29440R1

Dear Dr. Naheed Nazrul

We’re pleased to inform you that your manuscript has been judged scientifically suitable for publication and will be formally accepted for publication once it meets all outstanding technical requirements.

Kind regards,

Adu Appiah-Kubi, MBChB, CEMBA, FGCS

Academic Editor

PLOS ONE

Additional Editor Comments (optional):

Reviewers' comments:

Reviewer's Responses to Questions

**Comments to the Author**

1. If the authors have adequately addressed your comments raised in a previous round of review and you feel that this manuscript is now acceptable for publication, you may indicate that here to bypass the “Comments to the Author” section, enter your conflict of interest statement in the “Confidential to Editor” section, and submit your "Accept" recommendation.

Reviewer #3: (No Response)

2. Is the manuscript technically sound, and do the data support the conclusions?

Reviewer #3: Yes

3. Has the statistical analysis been performed appropriately and rigorously? 

Reviewer #3: Yes

4. Have the authors made all data underlying the findings in their manuscript fully available?

Reviewer #3: Yes

5. Is the manuscript presented in an intelligible fashion and written in standard English?

Reviewer #3: Yes

6. Review Comments to the Author

Reviewer #3: It is excellent how the writers have succinctly and clearly presented their findings. There is straightforward presentation and interpretation of the results in the methods section, which is written with ease of comprehension. The results are well-positioned within the larger body of research by the discussion section, and the conclusions are reasonable and solidly backed by the evidence. As a whole, the manuscript is skillfully composed and constructed.

7. PLOS authors have the option to publish the peer review history of their article (what does this mean?). If published, this will include your full peer review and any attached files.

Reviewer #3: **Yes: **Michael Yaw Amoh

---

## [Editor Report · Acceptance letter]

31 Jul 2024

PONE-D-23-29440R1 

PLOS ONE

Dear Dr. Nazrul, 

I'm pleased to inform you that your manuscript has been deemed suitable for publication in PLOS ONE. Congratulations! Your manuscript is now being handed over to our production team.

Kind regards, 

on behalf of

Dr. Adu Appiah-Kubi 

Academic Editor

PLOS ONE